# European and Mediterranean Myzocallidini Aphid Species: DNA Barcoding and Remarks on Ecology with Taxonomic Modifications in An Integrated Framework

**DOI:** 10.3390/insects13111006

**Published:** 2022-11-01

**Authors:** Giuseppe Eros Massimino Cocuzza, Giulia Magoga, Matteo Montagna, Juan Manuel Nieto Nafría, Sebastiano Barbagallo

**Affiliations:** 1Dipartimento di Agricoltura, Alimentazione e Ambiente, University of Catania, 95123 Catania, Italy; 2Dipartimento di Scienze Agrarie ed Ambientali (DISAA), Università degli Studi di Milano, Via Celoria 2, 20133 Milano, Italy; 3Dipartimento di Agraria, Università degli Studi di Napoli Federico II, Via Università 100, 80055 Portici, Italy; 4BAT Center–Interuniversity Center for Studies on Bioinspired Agro-Environmental Technology, University of Napoli Federico II, Via Università 100, 80055 Portici, Italy; 5Departamento de Biodiversidad y Gestión Ambiental, University of León, 24071 León, Spain

**Keywords:** Hemiptera Aphididae, *Myzocallis*, molecular analysis

## Abstract

**Simple Summary:**

*Myzocallis* is a Holarctic genus of monoecious species of aphids mostly hosted by plants belonging to Fagales. To date, extensively morphological studies have been carried out on this group of aphids, but only sporadic molecular studies have been performed to understand the relationships among the different species. With the aim improve knowledges on these aspects, almost all species of the European and Mediterranean Myzocallidini species were investigated. As a consequence of the results obtained, *Myzocallis* (*Agrioaphis*) *leclanti* originally described as a subspecies of *M*. (*A*.) *castanicola* and *M*. (*M*.) *schreiberi*, considered a subspecies of *M*. (*M*.) *boerneri,* should be regarded at a rank of a full species. Moreover, the subgenus *Agrioaphis*, *Lineomyzocallis, Neomyzocallis* and *Pasekia* were elevated to the rank of genus, while *Myzocallis* remain as such.

**Abstract:**

The genus *Myzocallis* Passerini (Hemiptera, Aphididae, Calaphidinae, Myzocallidini) is a rather primitive group of aphids currently comprising 45 species and 3 subspecies, subdivided into ten subgenera, three of them having a West Palaearctic distribution. The majority of the species inhabit Fagales plants and some of them are considered pests. Despite their ecological interest and the presence of some taxonomic controversies, there are only a few molecular studies on the group. Here, the main aims were to develop a DNA barcodes library for the molecular identification of West Palaearctic *Myzocallis* species, to evaluate the congruence among their morphological, ecological and DNA-based delimitation, and verify the congruence of the subgeneric subdivision presently adopted by comparing the results with those obtained for other Panaphidini species. These study findings indicate that *Myzocallis* (*Agrioaphis*) *leclanti,* originally described as a subspecies of *M*. (*A*.) *castanicola* and *M*. (*M*.) *schreiberi*, considered as a subspecies of *M*. (*M*.) *boerneri,* should be regarded at a rank of full species, and the subgenera *Agrioaphis, Lineomyzocallis, Neomyzocallis*, *Pasekia* were elevated to the rank of genus, while *Myzocallis* remain as such.

## 1. Introduction

The Holarctic genus *Myzocallis* Passerini (Hemiptera, Aphididae, Calaphidinae, Myzocallidini) currently lists 45 species, including a few subspecies [1,2]. It is subdivided into ten subgenera, three of them having a West Palaearctic native area of distribution as quoted in the revision of all species-group on a worldwide scale [3,4]. Except for the Nearctic *M. asclepiadis* (Monell) living on *Asclepias* (Apocynaceae), all other *Myzocallis* species inhabit plants belonging to Fagales. Most species live on host plant of the genera *Quercus* and *Castanea* (Fagales, Fagaceae), except for *M. myricae* (Kaltenbach) linked to *Myrica* (Myricales, Myricaceae), and *M. carpini* (Koch) and *M. coryli* (Goeze) living on the Betulaceae genera *Carpinus* and *Corylus*, respectively [1]. Compared to the more recent and numerous Aphidinae, Calaphidinae aphids are rather primitive, according to the hypothesis they had a parallel evolution with Fagales, an earlier differentiated group among Magnoliophyta or Angiospermae [4]. In a recent taxonomic revision integrating data from molecular and morphological analyses, the west Eurasian oaks have resulted inclusive of subgenus *Quercus*, with the sections ‘Quercus’ and ‘Ponticae’ and of subgenus *Cerris*, in turn subdivided into Sections ‘Cerris’ and ‘Ilex’ [5]. Noteworthy, each European oak taxa host at least one Calaphidinae aphid species.

All known species of *Myzocallis* were keyed and finely illustrated by [4]. Further papers on taxonomy, morphology, distribution and host plants of European *Myzocallis* species are those of [6,7,8,9]. Recently, some species of the genus *Myzocallis* have been also included in a molecular study on the subfamily Calaphidinae [10].

The three West Palaearctic subgenera of *Myzocallis* (i.e., *Agrioaphis*, *Myzocallis* and *Pasekia*) are represented so far by thirteen species and one subspecies [3]. One of these species, *M*. (*M*.) *macrolepidis*, was recently described from Italy [11]. Four of these taxa (*viz*. *M. castanicola* Baker s. str., *M. carpini*, *M. coryli* and *M. boerneri*) are found also outside the native area since they were introduced into other continents following the human activities [1,4]. In contrast, one species of the large Nearctic subgenus *Lineomyzocallis*, *M. walshii* (Monell) appeared in Europe around the end of the ‘80s and quickly became widespread on the introduced red oak, *Q. rubra* (subgenus *Quercus*, sect. Lobatae) [8,12,13]. The Mediterranean *Myzocallis* are monoecious and predominantly holocyclic, except for of *M. schreiberi* Hille Ris Lambers & Stroyan and *M. cocciferina* Quednau & Barbagallo, which develop anholocyclically on evergreen oaks. All the species live on the lower part of the leaves usually without causing appreciable damage and are not myrmecophilous. West Palaearctic *Myzocallis* species are characterized by small body size (1.3–2.6 mm) and body colour varying from pale straw yellow to ocherous. Some species dorsally show dark longitudinal strips on the head and thorax and dark spots on the abdomen, while in others those patches are barely visible. Moreover, the viviparous females of these species are characterized by knobbed cauda and bilobed anal plates. All the viviparous females of *Myzocallis* species are also alate, except for of *M. glandulosa* and occasionally *M. coryli*, for which apterous or apteroids forms are known. Nymphs, as well as apterous viviparous and oviparous females, have dorsal and sometimes basal antennal hairs rather long and capitate [1,7].

Within the *Myzocallis* subgenera, the species are sometimes rather difficult to be distinguished based on morphology, since only few diagnostic characters are present that are also frequently subject to biometric variations among the different populations due to the influence of various abiotic factors [4]. These species groups have been extensively studied from the morphological point of view, but only a few molecular analyses were performed on them. DNA barcoding method [14] represents a useful tool for insect taxonomy since, in most cases, it allows discrimination of species based on molecular information even when their morphological identification is difficult [15,16,17,18,19]. The method has been used in numerous researches carried out on various systematic groups of aphids for the identification of the species [20,21,22], to associate different morphs and hosts [23,24,25], to recognize crop pest species [26] and the invasion history of pest species [27]. Beyond the identification of species, DNA barcoding frequently allows highlighting inconsistencies between morphological and molecular species identification and DNA barcodes have been proved to be effective also in species delimitation [28,29].

The main aims of this research are (*i*) to develop a DNA barcodes library for the molecular identification of Mediterranean species of the genus *Myzocallis*; (*ii*) to evaluate the congruence among morphological, ecological and DNA-based delimitation of the taxa belonging to this genus, benefitting also of comparison with molecular data of other Panaphidini species; and (*iii*) verify the congruence of the subgeneric subdivision presently adopted.

## 2. Materials and Methods

### 2.1. Aphids Collection and Specimen Identification

The study was carried out on about 400 samples representatives of almost all *Myzocallis* species present in the Mediterranean area plus other closely related Panaphidini species (*Apulicallis trojanae* Barbagallo & Patti, *Tuberculatus eggleri* Börner, *T. neglectus* Krzywiec, *T. remaudierei* Nieto Nafría, *Hoplocallis ruperti* Pintera and *H. picta* Ferrari, *Siculaphis vittoriensis* Quednau & Barbagallo). The specimens were dropped on the surface of a wooden plate by beating the leaves and then placed in tubes containing 85% alcohol. The leaves and acorns of each plant from which the aphids were collected were taken and stored for subsequent recognition. The host plants on which the analysed individuals were collected were identified following most accredited European Flora handbooks [30,31]. Metadata on the analysed specimens, such as host plants, geographical coordinates, sampling date and GenBank accession, are reported in Appendix A.

For each sample collected, a preliminary classification of individuals was carried out through observation under the microscope. Subsequently, to confirm the first visual morphological classification a number of specimens were mounted on slides, according to the current preparation method for aphids and particularly for the softest Calaphidinae species [32,33]. The identification was carried out using characters reported in the key [4] and the comparison to specimens in the collection of senior co-authors (S.B. and J.M.N.N.). One single alate viviparous has been seen for *M. persica* Quednau & Remaudière and also *M. taurica* Quednau & Remaudière; moreover, few specimens of *M. glandulosa* Hille Ris Lambers (either as alate and apterous viviparous females) have been seen, thanks to the courtesy of Mr. Paul Brown—The Natural History Museum of London. Slides are available in the co-authors’ collections (Dept. of Agri-Food and Environmental Systems Management, University of Catania, Italy and the Dept. of Biodiversity and Environmental Management, Univ. of León, Spain). The largest part of the collected specimens was then stored in 95% (−20 °C) ethanol for the subsequent molecular analyses.

### 2.2. DNA Extraction, Amplification and Sequencing

After the morphological examination under the stereo-microscope, 3–4 samples, representative of each sample, were randomly selected for the molecular analysis. Total DNA was extracted from single individuals using the DNeasy Blood & Tissue kit (Qiagen, Hilden, Germany) following the instruction suggested by the manufacturing company. The non-destructive method [10] was used, so to not preclude subsequent morphological analyses of the specimen if molecular analysis makes it necessary. A fragment of mitochondrial COI was amplified using the universal primers LCO1490 and HCO2198 [34]. All PCRs were performed in 10 µL, with 4.25 µL buffer premix 2 ×
F (FailSafe tm PCR Premix Selection Kit, Epicentre Technologies, Thane, India), 0.5 µL of each primer (10 pmol, 0.25 µL Taq polymerase (Life Technologies, Carlsbad, CA, USA) and 2 µL DNA template. Thermal PCR cycle and electrophoresis conditions see [25]. PCR products were sent for sequencing to BMR Genomics (Padua, Italy) using ABI PRISM 3730XL DNA sequencer. All chromatograms were evaluated using 4Peaks [35], low-quality sequences were excluded from the following analyses and doubtful initial or final regions were pruned. The presence of open reading frame was assessed in order exclude nuclear mitochondrial pseudogenes. The developed sequences were deposited in Genbank (accessions list in Appendix A).

### 2.3. Nucleotide Distance Analyses and Taxa Molecular Delimitation

All COI sequences of the Panaphidini subfamily publicly available on BOLD [36] were retrieved, and then aligned together with those developed in this study. Alignment was performed using MUSCLE algorithm [37] implemented in MEGA X [38]. Alignment was trimmed to retain the region shared among the majority of the sequences (616 bp segment within Folmer region [34] and then all the sequences with length <390 bp were excluded. R library Haplotypes (https://biolsystematics.wordpress.com/r/ accessed on 18 March 2022) was used for reducing haplotypes within each species. Finally, some of the sequences retrieved from BOLD were removed from the dataset or their identifier modified for the following reasons: incompatibility between the individual morphological identification and the collection host plant; sequences related to misidentifications (previously signalled by other scientific works) (Appendix A). This dataset was then split into two sub-datasets, i.e., Myzocallis genus sub-dataset and a dataset including all Panaphidini sequences except for Myzocallis ones, in order to perform the analyses described hereafter.

From the Panaphidini dataset, the sequences belonging to the Myzocallis genus were extracted using R software and a Kimura-two parameter (K2P) [39] pairwise nucleotide distance matrix was estimated starting from them using ape R library [40]. The obtained nucleotide distance matrix was analysed for extrapolating summary statistics on intraspecific and interspecific distances, intrasubgeneric and intersubgeneric distances, and intrageneric distances using the R library spider [41].

Species delimitation analyses were performed on the same Myzocallis nucleotide sequences dataset using two species delimitation methods (i) 2% nucleotide distance threshold as species clustering threshold, a value corresponding to the maximum intraspecific distance estimated by [23] on aphids COI sequences. This analysis was carried out using the R package spider [41]. (ii) Assemble Species by Automatic Partitioning Estimation (ASAP) [42]. The delimitation was performed on the server (https://bioinfo.mnhn.fr/abi/public/asap) using the K2P model [39] and the remaining parameters were set as default. ASAP delimitation was defined by evaluating both the partitions with the first and the second best asap-score.

The previously aligned COI gene sequences of the genus Myzocallis were used to infer a single-gene phylogenetic tree using both Bayesian and Maximum Likelihood approaches. The best nucleotide substitution model was estimated using PartitionFinder2 [43] and selected according to the Bayesian information criterion (BIC) [44]. According to BIC, the best model of nucleotide substitution resulted the HKY model [45] with gamma distribution (Γ) and proportion of invariable sites (I). Bayesian inference was performed using MrBayes 3.2.2 [46] with two independent runs of 3 × 10^7^ generations (sample frequencies: every 100 generations; stationarity reached when the average standard deviation of split frequencies <0.01) and the nucleotide substitution model settled according to the results of the model selection analysis. The convergence of the runs was visually inspected using TRACER [47] and an appropriate number of sampled trees were discarded as burn-in. The Maximum Likelihood inference was performed using PhyML 3.0 [48] implementing: the selected model of nucleotide substitutions; tree searching operations accounting for the best between the nearest neighbour interchange and the subtree pruning and regrafting; approximate Likelihood-Ratio Test (aLRT) [49] as branch support.

The sub-dataset of Panaphidini sequences not including Myzocallis genus was created excluding the sequences of the Myzocallis genus from the full Panaphidini dataset using R software. This dataset was used for estimating a K2P pairwise nucleotide distance matrix from which intergeneric distances were derived. From the same dataset, the sequences of genera represented from at least two species were extracted and used for the estimation of K2P intrageneric distances.

## 3. Results

In this study, a DNA barcode library including sequences from 63 Panaphidini individuals was developed. Processed individuals belonged to 19 species, 13 of them of the genus *Myzocallis*, while the remaining of the genera *Tuberculatus* (three species), *Hoplocallis* (one species), *Apulicallis* (one species) and *Siculaphis* (one species) (Appendix A). The mean length of the obtained barcode sequences was of 616 bp [range: 600–616 bp] with the following average base composition: A = 34.6%, C = 15.1%, T = 40.5%, G = 9.8%.

### 3.1. Nucleotide Distance Analyses

The *Myzocallis* COI sequences developed in this study plus all the sequences available for this genus in BOLD, for a total of 90 sequences belonging to 17 species (representative of five different *Myzocallis* subgenera, i.e., *M. Agrioaphis*, *M. Lineomyzocallis*, *M. Myzocallis*, *M. Neomyzocallis*, *M. Pasekia*) were assembled in a dataset that includes ~43% of the species currently described for this genus.

Nucleotide divergence within and among *Myzocallis* species resulted in a mean value of 2.1% (range: 0.2–11.4%) and 10.4% (range: 0–18.3%) respectively, while intrageneric divergence was estimated to be in mean of 9.6% (Figure 1a). *Myzocallis* subgenera intra- and inter-subgeneric nucleotide distances resulted to be in mean of 4.5% and 11.6%, respectively (Figure 1b,c). The highest inter-subgeneric nucleotide distances have emerged between subgenera *Myzocallis* and *M*. *Lineomyzocallis* (mean value 13.4%) and between the latter and *Pasekia* (mean value 11.9%). A notable situation emerged within the subgenus *Myzocallis*, where two groups showing a considerable nucleotide distance (in mean 8.1%, range: 6.1–10.7%) were recognised, one including *M. coryli*, *M. carpini*, *M. boerneri* and *M. occidentalis* (*M. Myzocallis* group A) and the other including *M. schreiberi*, *M. glandulosa*, *M. macrolepidis* (*M. Myzocallis* group B). When intra-/inter-subgeneric divergences were estimated considering these groups as two different subgenera, a mean value of 3.7% and 11.4%, respectively, were found (Figure 1d,e).

The dataset assembled in this study including all sequences of Panaphidini except for Myzocallis (de novo developed plus BOLD database publicly available barcodes) resulted to be composed of 580 COI sequences. Specifically, it included 118 species belonging to 41 genera with a mean intergeneric divergence between genera of 13.1% (Figure 1g) and mean intrageneric divergence, estimated only on genera represented by at least two species (520 sequences of 20 genera), of 6.9% (Figure 1f).

### 3.2. Species Delimitation of the Genus Myzocallis

The species delimitation analyses performed on the genus *Myzocallis* dataset using two molecular delimitation methods (i.e., the 2% clustering threshold and ASAP) produced comparable quite results and partially reflect the classical subdivision based on morphology. Specifically, 2% threshold delimitated 21 evolutionary units within the dataset, 11 of them exactly matched the morphological species. The species *M. walshii* and *M. castanicola* were both split in three evolutionary units, while all *M. carpini* and *M. coryli* were merged in the same one (except for two samples). Finally, also *M. asclepiadis* and *M. punctata* were merged in a single unit (Figure 2). Species delimitation adopting ASAP method led to almost identical results with the exceptions of *M. carpini* and *M. coryli* whose sequences were split into different evolutionary units but never merged together, and of *M. castanicola*, whose sequences were split in four units, one of them including only *M. leclanti* sequences (Figure 2).

In the *Myzocallis* genus dendrogram, inferred using the same COI sequences dataset, not all the species resulted monophyletic, with incongruences between specimens’ morphological identification and monophyly in tree involving some of the species already highlighted from species delimitation analyses (six species, Figure 2). Whereas, all *Myzocallis* subgenera, except for *Agrioaphis*, formed monophyletic clusters (nodes support aLRT ≥ 0.85, except for *Pasekia*; Figure 2). Noteworthy, the two groups identified within the subgenus *Myzocallis* from nucleotide distance analyses (i.e., subgen *Myzocallis* group A and *Myzocallis* group B) resulted in two well supported monophyletic groups (aLRT = 0.95 node support) (Figure 2).

## 4. Discussion

The analyses performed in this study shed light on some aspects of the taxonomy of the genus *Myzocallis*. The main taxonomic inferences derived from this study can be summarized as follow: (*i*) two different groups are present within the subgenus *Myzocallis* whose taxonomic rank should be better investigated; (*ii*) the nominal taxon *M. castanicola leclanti* Quednau & Remaudière possibly should be elevated to the level of full species; (*iii*) the status of valid species of *M. schreiberi* Hille Ris Lambers & Stroyan, which in the past was considered as a synonym of *M. boerneri* Stroyan, has been confirmed also based on molecular analyses; (*iv*) the taxonomic hierarchy of *Myzocallis* subgenera have to be revised, together with other related genera of Myzocallidini.

### 4.1. Myzocallis Subgenus Myzocallis

The analyses revealed that the subgenus *Myzocallis* is divided into two well separated clusters (in terms of nucleotide distance) based on COI gene sequences. The first cluster joints together *M. coryli, M. carpini*, *M. boerneri* and *M. occidentalis* (*Myzocallis* group A)*,* while the second includes *M. schreiberi*, *M. glandulosa*, and *M. macrolepis* (*Myzocallis* group B). All these species show common morphological characters that allow to cluster them as a single homogeneous group based on morphology. The observed genetic differences could be the result of an ecological divergence of the two species groups. The host plants of the species of *Myzocallis* group A are all deciduous, on the contrary, those of *Myzocallis* group B are all semi-evergreen. Aphids’ cycles have to fit the biology of the host plants, and in fact the aphids of the *M. Myzocallis* group B often develop anholocyclic on them. Except for *M. coryli*, all other species of *M. Myzocallis* group A showed a very low intraspecific nucleotide divergence based on COI, independently from their geographical origin (except for one sequence of *M. carpini* that will be further discussed). *M. coryli* was morphologically well-studied by several authors, but the presence of morphological variation among the populations of this species was never reported. However, for this species an unusually high intraspecific genetic variability was already observed (on the COI gene), especially between individuals from different geographical areas [10,50]. In previous researches, the authors hypothesized that *M. coryli* is a cryptic species complex. The nucleotide sequences of *M. coryli* analysed in this study (18 sequences in total, 14 of which were mined from BOLD) resulted subdivided into three clusters plus one independent sequence in the COI dendrogram and were differentially delimited by the two species delimitation methods adopted, i.e., two (clustering threshold) and three (ASAP) different evolutionary units (Figure 2). Intraspecific nucleotide distance of such a species ranged from 0.02% to 5%. Based on this evidence, *M. coryli* may represent a complex of cryptic species. Anyway, since *M. coryli* is a holocyclic and monoecious species on *Corylus* genus, the nucleotide distances and clusters observed could be explained by the wide geographic distribution of its host plant. Although the latter has a European-Caucasian origin, currently it is widespread throughout the world for agricultural production with *M. coryli* that followed its distribution. This may have determined a progressive adaptation of the aphid populations to the different environmental conditions and led to some genetic divergence between them, without evolving morphological differences and maybe nether incurring in speciation. This situation might have some analogy *B. helichrysi*, where a COI nucleotide distance of 2.7% was detected between two different populations and for which the existence of two cryptic species has been hypothesized [51,52]. However, further investigation within this group should be undertaken even with additional mitochondrial and nuclear markers.

The analyses performed in this study clearly distinguished the holm oak aphid *M.* (*M.*) *schreiberi* from the Turkey oak aphid *M*. (*M*.) *boerneri* Stroyan. Both species delimitation methods recognised them as separated species, but also, they have fallen into separate clusters in COI tree (*Myzocallis* group A, *Myzocallis* group B) (Figure 2). The two species are morphologically very similar, so in the past, they have often been confused, doubting whether they were distinct species or assuming that *M. schreiberi* was an anholocyclic form of *M. boerneri* [3,4,53]. This misunderstanding is probably at the origin of the incorrect classification of the COI sequences ACEA810-14 and GBMHH5717-14 available on BOLD (renamed as *M. schreiberi* for these study analyses; see Materials and Methods). While describing *M*. (*M*.) *schreiberi* from specimens collected on *Q. ilex* in Italy and England, Hille Ris Lambers & Stroyan [54] listed the morphological characters distinguishing this species from other European congeneric taxa known at the time. While the identification keys to distinguish *M. schreiberi* and *M. boerneri* are provided by Barbagallo & Massimino Cocuzza [11]. These two aphids have a rather different host plant preference. *M. schreiberi* usually lives on *Q. ilex* and on other evergreen—leaved oak species, such as *Q. suber* and *Q.* x *crenata*. All these oaks have coriaceous leaves with a grey-tomentose texture beneath, to which this aphid species is likely adapted to feed on through its rather acute shaped last rostral joint. In the present study, no genetic difference was found among the populations of *M. schreiberi* living on *Q. ilex*, *Q. suber* and *Q.* x *crenata*. *M. boerneri* is widely distributed in the West Palaearctic region (Europe, Middle East) and its main host plant is the deciduous *Q. cerris*.

The *M*. (*M*.) *glandulosa* Hille Ris Lambers [55] and *M*. (*M*.) *occidentalis* Remaudière and Nieto Nafria [56] are both characterized by a rather long last rostral joint. The former was described from *Q. ithaburensis* in the Middle East [48]. Several years later, the alate viviparous female of *M*. (*M*.) *glandulosa* was figured and *Q. persica* was added as a further host plant for this species [4]. The individuals whose COI sequences were analysed in the present study were collected in Israel from the type locality. *M*. (*M.*) *occidentalis* has been described from specimens collected in South-Western Europe (France, Spain); the only host plant known is *Q. pyrenaica*, on which the aphid performs a holocyclic life cycle [8,55]. Also included in this group is *M. macrolepidis*, a new species of recent description [11]. Moreover, the sequence labelled as *M. occidentalis* presents on NCBI (accession number GBMIN66582) obtained from a specimen collected in California on *Quercus* sp. [10] should be regarded as belonging to a different taxon and would deserve in-depth analysis. Finally, also the classification as *M. occidentalis* of the samples caught with suction or yellow water traps in Serbia and Greece could be incorrect [57,58].

### 4.2. Myzocallis Subgenus Agrioaphis

The subgenus *Agrioaphis* Walker is represented so far by two species *Myzocallis myricae* and *M. castanicola* (the latter including two subspecies, *M. castanicola castanicola* and *M. castanicola leclanti*) [3]. *M. castanicola* mainly lives on *Quercus*, but it was recorded as well from *Castanea* by several authors. The very common populations living on *C. sativa* in southern Europe and the Middle East, usually belong to the subspecies *M. castanicola leclanti*. On chestnut it performs a monophagous and holocyclic life cycle and not rarely is considered a noxious aphid species. In Italy, *M. castanicola castanicola* is quite common in northern areas of the peninsula, mainly on *Q. petraea* and its hybrids x *Q. pubescens* (group) and unlikely it can be found there on different *Quercus*-species (including *Q. robur*) or on *C. sativa*, unless perhaps as occasional vagrant alates. In Spain, the aphid commonly lives on *Q. pyrenaica* and it is occasionally detected on additional oak species [8].

Based on the results obtained in the present study, the bog myrtle aphid *M. myricae* (Kaltenbach) is clearly distinguishable from *M. castanicola*. Interestingly, *M. castanicola* was split into three or four evolutionary units based on the delimitation methods adopted (Figure 2). These units appear to be related with the host plant on which individuals were collected. In the case of ASAP, the evolutionary units were composed as follow (*i*) individuals collected from *Q. pyrenaica* and *Q. cerris* (plus two sequences mined from BOLD for which the host plants are not specified), (*ii*) individuals collected from *Q. petraea* in North Est Italy, (*iii*) individuals collected on *Castanea sativa* and currently classified as *M. castanicola* subsp. *leclanti,* (*vi*) a sole individual whose sequence was mined from BOLD (GBMHH16898-19). A 2% delimitation threshold partially confirmed these results, but units (*ii*) and (*iii*) were merged together since the nucleotide distance between individuals is ~2%. Further analyses are needed to shed light on the taxonomic status of the individuals following in these clusters, but we hypothesis *M. castanicola* GBMHH16898-19 is a misidentification (actually this sequence may belong to a species not present in the dataset of this study). On the other hand, the analyses here performed clearly discriminate *M. castanicola castanicola* and *M. castanicola leclanti*, as belonging to two separate evolutionary units. *M. castanicola castanicola and M. castanicola leclanti* also clearly differ from the morphological point of view. Specifically, *M. castanicola leclanti* has (i) paler yellow colour of nymphs, alate viviparous females and sexuales; (ii) paler and less extensive dorsal sclerified areas, particularly those on abdomen; (iii) more haired last rostral joint. Quednau & Remaudière [3] and Quednau, [4] provided the morphological description and illustration of *M*. *castanicola leclanti*, adding the comparative range of their variation within the different seasonal morphs of *M*. *castanicola castanicola*.

### 4.3. Myzocallis Subgenus Pasekia

The subgenus *Pasekia* Aizenberg is represented so far by five species, i.e., *Myzocallis persica*, *Myzocallis taurica, Myzocallis komareki, Myzocallis mediterranea* and *Myzocallis cocciferina*. All of them are well studied from the morphological point of view and a key for their identification (as viviparous alates and nymphs, including males) has been reported [3,4].

*M.* (*Pasekia*) *persica* Quednau & Remaudière and *M*. (*P*.) *taurica* Quednau & Remaudière are apparently confined to eastern Mediterranean. Both species are holocyclic, the former on *Q. persica*, and the latter on *Q. coccifera* (maybe also other oak species). Unluckily, these species were not included in the present study since no specimen was available for the molecular analyses.

*M*. (*P*.) *komareki* Pašek has a wider distribution, it was recorded from Central Europe southward to the Mediterranean and the Middle East on several oak species, as well as on *Castanea sativa* [1,3,8,59]. The nucleotide distance analysis of this study showed low intraspecific variability (<2.1%) for *M. komareki*, despite the analysed specimens being collected from different oak species and various geographical localities. It can be argued that, due to its polyphagy, this species can easily adapt to the different oak species, but without occurring in isolation of its populations.

*M*. (*P*.) *mediterranea* Quednau & Remaudière is recorded so far from France, Spain and Italy, and probably it is widespread also in other Mediterranean countries. The species inhabits several oaks such as *Q. pubescens* and strictly allied taxa (Q. *congesta, Q. virgiliana* and Q. *dalechampii*) [60] and only occasionally it has been collected on oaks of different groups (i.e., *Q. ilex*). In the past, this aphid was frequently confused with *M*. (*P*.) *komareki*, because of their morphological similarity [61,62]. Usually, *M*. (*P*.) *mediterranea* develops through a regular holocycle. Nevertheless, in more warm habitats (such as in southern Italy) the amphigonic morphs appear very late in the season (from the end of December to the first half of February), inhabiting the host plant leaves still green in *Q. pubescens*-group species (semi-evergreen oaks). In such a case, part of the aphid population can overwinter anholocyclically on these oaks and the survived specimens move to new oak blossoms by the end of March or early in April [63] (as *M. komareki*). Also, in the case of *M. mediterranea*, a low intraspecific variability (<1% nucleotide distance) was found between specimens considered in this study, however all of them were collected from a limited geographic area (Sicily) on *Q. pubescens*.

Finally, *M*. (*P*.) *cocciferina* Quednau & Barbagallo is present in southern Europe and Mediterranean countries, from Spain eastward to Lebanon on *Q. coccifera s. lat*. and occasionally on *Q. ilex* or their hybrids (*ilex* x *coccifera*). In Spain, the aphid is recorded on *Q. ilex* and reported as well for Portugal and North Africa [8]. This species was originally attributed [64] to subgenus *Agrioaphis sensu* Richards (1968) and later transferred to *Pasekia* [3,64]. In the present study, molecular and morphological species delimitation were consistent for *M*. *cocciferina*. A nucleotide distance of ~1.5% was estimated between specimens collected from *Q. coccifera* in Sicily and Apulia and those collected from *Q. ilex* or its hybrid x *Q. coccifera* (postulated as *Q. soluntina* Tineo ex Lojacono by Giardina et al. [65]).

### 4.4. Nearctic Myzocallis Subgenera Lineomyzocallis and Neomyzocallis

In this study, a few specimens belonging to the Nearctic subgenera *Lineomyzocallis* (*M. walshii*, *M. bellus* and *M. ephemerata*) and *Neomyzocallis* (*M. asclepiadis* and *M. punctata*) were included. Here, two relevant situations were observed: (i) a high intraspecific variability of *M. walshii* (range intraspecific nucleotide distance: 0.2–8%); (ii) a very low nucleotide distance between *M. asclepiadis* and *M. punctata* (<0.2%). All the species delimitation methods adopted split *M. walshii* into three evolutionary units, two of them including one individual each (Figure 2). The black-bordered oak aphid, *M. walshii* lives on *Quercus* of the group of red oaks (Quercus section Lobatae) and is native to North America, where it is widely distributed. In 1989 the species was found for the first time in France and in about twenty years it spread throughout the European continent [1]. None of the numerous studies performed in the last decades on *M. walshii* reported the presence of intraspecific morphological or biological variability. Possibly, the high genetic divergence observed in this study could be related to specimen misidentifications, rather than to the existence of cryptic diversity. Both the molecular species delimitation methods used in this study delimited *M. asclepiadis* and *M. punctata* in a single species, in accordance with what was already observed [10], i.e., some sequences present on BOLD may have been mistakenly attributed to *M. asclepiadis*.

## 5. Conclusions

Systematics is a hierarchic science fundamentally built on the relationships of affinity between species [66]. COI is a DNA marker known to be effective in discriminating insect taxa at the lowest taxonomic levels. Combining molecular (COI gene), morphological and ecological information in an integrative framework makes possible to resolve numerous taxonomic issues concerning insect species [67,68,69,70,71]. In particular, the most accurate results are obtained when expert morphologists on the target group are involved in the evaluation of the signal resulting from molecular analyses, in this case is also easier to distinguish between extrinsic and intrinsic errors related to the species molecular identification and delimitation [71,72,73]). For some insect groups, as in the case of aphids, the evaluation of the ecological information is important in the process of species identification and delimitation as well. Hence the relevance of coupling DNA sequences with specimen’s collection locality and date, and further ecological information if present (e.g., collection habitat or host plant), even in the phase of molecular data publication.

However, also in this context, the use of integrative taxonomy (and COI as DNA marker) to define other taxonomic categories than species could sound as improper. Whereas, in previous studies it was done and valuable results were obtained. For example, the COI nucleotide distances have been examined for evaluating the range of variation between genera and subfamilies of Aphididae [10,23,27]. Lee et al. [10] developed the barcode sequences for 154 Aphididae species (72 genera, 11 subfamilies), and reported a mean pairwise divergence between specimens of different genera of 8.9% (range from 1.6 to 19%). Other studies, on Adelgidae and Eulachnini aphids [23], found a mean intergeneric distance of 9.6% and 11.7%, respectively. In a further investigation, a mean intergeneric distance of 7.7% (from 5 to 9.7%) in the tribe of Macrosiphini and 10.4 (range 8.9% to 12.4%) in Aphidini were found [27].

The *Myzocallis* subgenera analysed in this work are morphologically distinguishable through characters of well-verified validity [4], recognized as adequate by aphid-taxonomists. In the present study, a substantial nucleotide distance was found between the *Myzocallis* subgenera (Figure 1), reinforcing the validity of these taxa, regardless of their hierarchical position. The nucleotide distance values are sufficient to justify the elevation of the subgenera to the rank of genus. In particular, considering that the inter-subgeneric distances estimated in this study for *Myzocallis* subgenera are comparable to those observed between genera of the Panaphidini tribe (Figure 1), that certainly include species phylogenetically more distant than those of the *Myzocallis* genus.

Several authors stated that different genera have to be monophyletic [66,74], and the conclusion drawn by the present work meets this condition. Furthermore, this evidence could be supported also from the morphological point of view. Some of the most important morphological characters [3,4] (i.e., spinal hairs on abdominal tergites, chaetotaxy of immature morphs and oviparae) to distinguish *Myzocallis* subgenera from each other, are the same that were used to distinguish the genera *Hoplocallis* (until a few years ago considered as a subgenus of *Myzocallis*), *Apulicallis* and *Siculaphis*.

Consequently, we elevated these subgenera to the rank of genus: *Agrioaphis* Walker, **stat. n.** [described as genus, type species *Aphis myricae* Kaltenbach], *Lineomyzocallis* Richards, **stat. n.** [described as subgenus of *Myzocallis*, type species *Aphis bella* Walsh], *Neomyzocallis* Richards **stat. n.** [described as subgenus of *Myzocallis*, type species *Callipterus punctatus* Monell], and *Pasekia* Aizenberg **stat. n.** [described as subgenus of *Myzocallis*, type species *Hoplocallis komareki* Pašek, 1953]. The genus *Myzocallis* remains as such, including the nominotypical genus and the subgenera whose species have not been considered in this work: *Californicallis* [3], *Castaneomyzocallis* [3] Quednau & Remaudière, *Globulicaudaphis* Hille Ris Lambers, *Neodryomyzus* Quednau & Remaudière, *Neodryomyzus* Quednau & Remaudière, and *Paramyzocallis* Quednau & Remaudière, for which further studies are needed.

Finally, *Myzocallis* (*Agrioaphis*) *leclanti* Quednau & Remaudière, **n. stat**., originally described as a subspecies of *M*. (*A*.) *castanicola* (Quednau & Remaudière, 1994) and *M*. (*M*.) *schreiberi*, considered as a subspecies of *M*. (*M*.) *boerneri*, should be regarded at a rank of full species.

## Figures and Tables

**Figure 1 insects-13-01006-f001:**
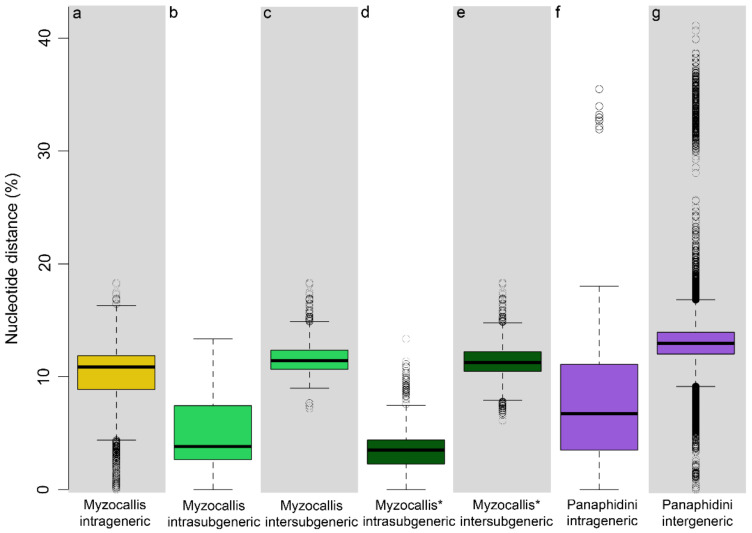
Boxplot of K2P pairwise nucleotide distances (**a**) within the genus *Myzocallis*; (**b**) within *Myzocallis* subgenera (**c**) between *Myzocallis* subgenera (**d**) within *Myzocallis* subgenera considering *M. Myzocallis* A and *M. Myzocallis* B as two separated subgenera (**e**) between *Myzocallis* subgenera considering *M. Myzocallis* A and *M. Myzocallis* B as two separated subgenera (**f**) within Panaphidini genera, excluding *Myzocallis* (**g**) between Panaphidini genera, excluding *Myzocallis*.

**Figure 2 insects-13-01006-f002:**
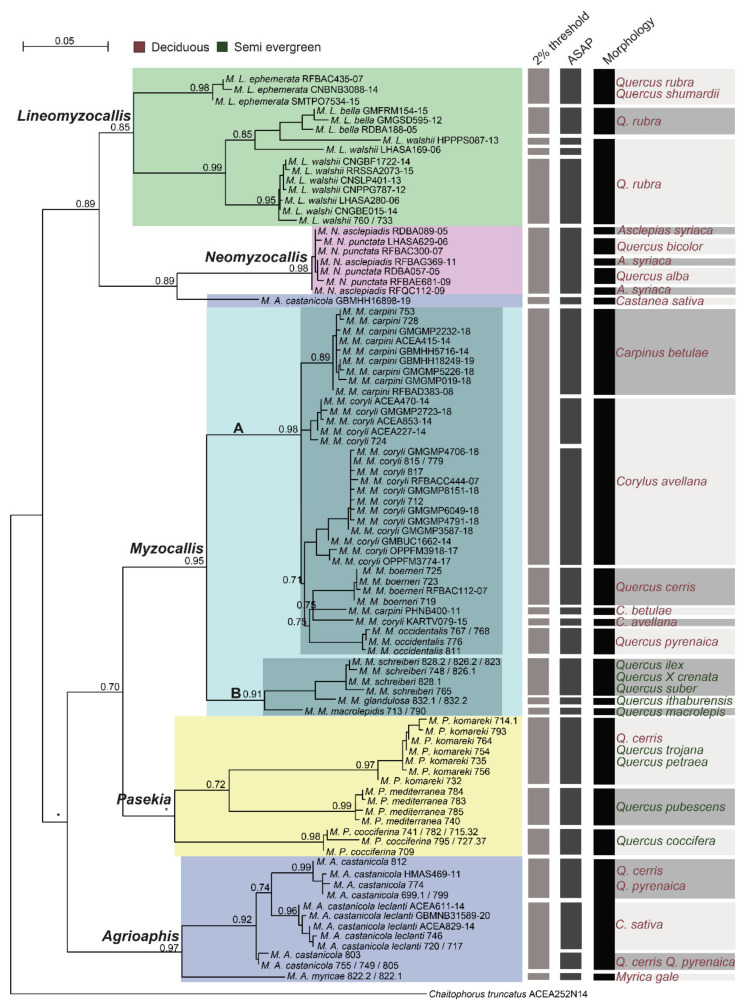
*Myzocallis* genus COI dendrogram and molecular species delimitation results. Results of molecular species delimitation analyses (i.e., 2% distance threshold and ASAP) and morphological delimitation are reported through vertical bars on the right side of the tree. On the same side, *Myzocallis* species host plants are reported (deciduous are written in red, semi evergreen in green), information on the host plants of species of the subgenera *M. Lineomyzocallis* and *M*. *Neomyzocallis* was taken from the scientific literature. BOLD id and identifiers of sequences developed in this study are indicated on the tips. On nodes aLRT values are reported, * represents aLRT values < 0.70. The tree scale bar indicates the distance in substitutions per site.

## Data Availability

Data presented in this research are available in the article.

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
