# Peer review of "European and Mediterranean Myzocallidini Aphid Species: DNA Barcoding and Remarks on Ecology with Taxonomic Modifications in An Integrated Framework"

_insects, 2022, doi:10.3390/insects13111006_

Round 1

Reviewer 1 Report

A good and useful paper. Please note a very few suggested language changes indicated on the attached copy of the manuscript

Author Response

Dear Reviewer, all the suggested linguistic corrections have been added. Thant you for your contribute to improve the manuscript. Sincerely, GMC

Reviewer 2 Report

The manuscript entitled “European and Mediterranean Myzocallidini aphid species: DNA barcoding and remarks on ecology with taxonomic modifications in an integrated framework” by Massimino Cocuzza et al. presents an interesting contribution to systematics of genus Myzocallis. The manuscript is well written and fluent and I suggest you to have it published after some minor corrections that I report below:

Line 50: You write that Calaphidinae aphids are rather primitive, there is some phylogenetic evidence that you can mention here?

Line 156: In the Nucleotide distance analyses and taxa molecular delimitation paragraph you work on two different dataset, the Myzocallis one and the Panaphidini one. I think that you should explain better how your dataset is built and describe from the beginning of the paragraph how you are going to split it in two different subsets.

Line 305: here the bibliography is reported for extended and not in numbers.

Line 463: I think it could be interesting here to discuss how the integration of morphological and molecular taxonomic identification can identify extrinsic errors of DNA barcoding (i.e. those relative to the quality of the reference dataset; i.e. the one you report in line 329 or in line 453) and intrinsic ones (i.e. those due to all those biological processes that generate a mismatch between mtDNA groups and species boundaries). The discrimination of this two kinds of errors can improve both the molecular identification tool and the knowledge of evolutionary history of this group of aphids. (see Salvi et al., 2020 https://doi.org/10.1371/journal.pone.0233573 and Marconi et al., 2022 https://doi.org/10.3390/d14080632)

Author Response

Dear Reviewer, all the suggestions and comments have been added to the text (in red in the text). Many thanks for your help as they improved the manuscript text. Best regards. GMC

Reviewer 3 Report

I found the manuscript very interesting. The Authors presented their results in very comprehensive but still very clear form. I am a great fan of integrative taxonomy approach and I think that we should use all possible methods of species identification and delimitation, especially in ambigious cases. I put my comments in the pdf file. The main concern I have is that in the case of groups of not fully solved taxonomy, Authors should be more careful with the conclusions. Additional mitochondrial and nuclear markers should be used to revise the problematic parts. At this point I would reard them as preliminary results as I mentioned in the text. I also suggest linguistic revision and checking the reference citation.

Author Response

Dear Reviewer, many thanks for your suggestions and comments that improved the manuscript. We added all the requested modifications (in red in the text). I agree with you that not all cases have been resolved. Investigations are continuing. Obviously, in the case of M. coryli it is necessary to collect a considerable number of specimens from different geographic origins and to use other molecular methods of investigation. The works are in progress. Many thanks. Sincerely, GMC

Reviewer 4 Report

Congratulations on your study! The development of a DNA barcodes sequence database is very important for all insect species. All suggestions for taxonomic changes seem to be well supported by the molecular characters evaluated. Just a just a suggestion to use more recent references.

Author Response

Dear Reviewer, Many thanks for your comment and suggestion. We have fixed something in the text and added two more citations of more recent articles.

Many thanks. Sincerely, GMC
